# A Randomised, Comparative, Effectiveness Trial Evaluating Low- versus High-Level Supervision of an Exercise Intervention for Women with Breast Cancer: The SAFE Trial

**DOI:** 10.3390/cancers14061528

**Published:** 2022-03-16

**Authors:** Rosalind R. Spence, Carolina X. Sandler, Benjamin Singh, Jodie Tanner, Christopher Pyke, Elizabeth Eakin, Dimitrios Vagenas, Sandra C. Hayes

**Affiliations:** 1Menzies Health Institute Queensland, Griffith University, Brisbane, QLD 4222, Australia; c.sandler@westernsydney.edu.au; 2School of Health Sciences, Western Sydney University, Campbelltown, NSW 2560, Australia; 3Kirby Institute, UNSW Sydney, Kensington, NSW 2052, Australia; 4Institute of Health and Biomedical Innovation, Queensland University of Technology, Kelvin Grove, QLD 4059, Australia; benjamin.singh@connect.qut.edu.au (B.S.); jm.tanner@connect.qut.edu.au (J.T.); 5Mater Health Services, South Brisbane, QLD 4101, Australia; christopher.pyke@mater.org.au; 6Mater Clinical Unit, Faculty of Medicine, The University of Queensland, South Brisbane, QLD 4101, Australia; 7School of Public Health, Faculty of Medicine, The University of Queensland, Herston, QLD 4006, Australia; e.eakin@uq.edu.au; 8School of Public Health and Social Work, Faculty of Health, Queensland University of Technology, Kelvin Grove, QLD 4059, Australia; dimitrios.vagenas@qut.edu.au

**Keywords:** exercise, neoplasms, survivorship, safety, patient compliance

## Abstract

**Simple Summary:**

The compelling evidence demonstrating the benefits of exercise to cancer survivors is biased towards ‘more well’ patients undertaking exercise interventions in tightly controlled (highly supervised) conditions. The aim of this trial was to evaluate the safety, feasibility, and effect of a 12-week exercise intervention delivered under ‘real-world’ conditions (that is, with low-level supervision, defined as five sessions across 12 weeks) compared with the same exercise intervention delivered under high-level supervision (20 sessions across 12 weeks). The understudied, less well women with breast cancer were the target population; that is, women with stage II or higher disease at diagnosis, and at least one comorbidity or treatment-related side-effect. The results showed that exercise was safe and feasible for this understudied breast cancer subgroup, and, while women who received the exercise intervention with low-level supervision experienced improvements in quality of life and physical function, greater gains in strength and exercise self-efficacy were observed for women who had the exercise intervention delivered via high-level supervision. Future research will determine whether the extra benefit gained through higher supervision levels lead to longer term quality of life and survival benefits.

**Abstract:**

The aim of this comparative, effectiveness trial was to evaluate the safety, feasibility and effect of an exercise intervention delivered via low-level versus high-level supervision. The target population were women who were diagnosed with ≥stage II breast cancer, had ≥ one comorbidity and/or persistent treatment-related side-effects, and were insufficiently physically active. Sixty women (50 ± 9 years) were randomized to the low-supervision group (*n* = 30) or high-supervision group (*n* = 30). The low-supervision group participated in a 12-week, individually-tailored exercise intervention supported by five supervised sessions with an exercise professional. The high-supervision group participated in the same exercise intervention but received 20 supervised sessions across the 12-week period. The target weekly dosage of 600 metabolic equivalent minutes of exercise per week (MET-mins/wk) and the session content, such as safety and behaviour change topics, were standardized between the groups. The primary outcomes were intervention safety, defined as the number, type, and severity of exercise-related adverse events (e.g., musculoskeletal injury or exacerbated treatment-related side effects), and feasibility, which was defined as compliance to target exercise dosage. The effect of the intervention on quality of life, physical activity, self-efficacy, fitness, and strength was also assessed (pre- and post-intervention, and at 12-week follow-up). The intervention was safe, with no exercise-related adverse events of grade 3 or above in either group. Both groups reported high compliance to the target exercise dosage (median MET-mins/wk: High = 817; Low = 663), suggesting the exercise intervention was feasible, irrespective of supervision level. Improvements in quality of life, physical activity and fitness were observed post-intervention and maintained at follow-up for both groups (*p* < 0.05). Only the high-supervision group showed clinically-relevant improvements in strength and self-efficacy at post-intervention (*p* < 0.05). Individually-targeted exercise delivered under high- or low-levels of supervision is safe, feasible and beneficial for women with stage II+ breast cancer. Future research needs to assess whether the greater gains observed in the group who received higher supervision may contribute to longer term maintenance of physical activity levels and overall health benefits. Australian and New Zealand Clinical Trials Registry: ACTRN12616000547448.

## 1. Introduction

The compelling evidence-base demonstrating the benefit of exercise for cancer survivors supports the continued call for exercise to be formally integrated into care during and following treatment [1]. This evidence has informed internationally-endorsed exercise guidelines [1,2], including specific weekly exercise dosage recommendations to improve physical functioning, health-related quality of life, anxiety, depression, and fatigue [1]. International exercise oncology guidelines recommend exercise dosages that approximate those recommended for healthy adults; that is, 150 min of moderate-intensity exercise per week [1]. However, as highlighted by these guidelines, confidence regarding the safety, feasibility, benefit and generalizability of the extant exercise oncology evidence is limited by (1) a bias towards samples of patients with the most common cancers and better prognoses, and (2) a lack of evaluation of delivery models suited to resource-constrained health care systems [1,2,3,4,5].

Compared with the wider breast cancer population, participants in exercise and breast cancer trials are more likely to have earlier-stage disease, higher pre- and post-diagnosis physical activity (PA) levels, and be otherwise generally well (that is, have few or no comorbidities, such as arthritis or pre-existing cardiovascular disease, and have mild or no persistent treatment-related side-effects) [5,6]. Higher stage of disease at diagnosis has been associated with higher frequency and severity of treatment-related side-effects and lower levels of PA [7]. It therefore seems plausible that those who have the most to gain from exercise are less likely to have participated in exercise oncology research, and concurrently, may be at higher risk of exercise-related adverse events and/or may not be able or willing to participate, although this topic remains understudied [8,9].

The majority (>60%) of published exercise oncology studies are efficacy trials evaluating exercise interventions conducted under highly controlled conditions [10]. These trials involved highly-supervised exercise interventions, with at least one exercise session per week supervised by an exercise professional (ExP) with advanced training in exercise oncology [10]. Additionally, these trials were primarily conducted at university or hospital clinics rather than within community-based settings [10]. In contrast, effectiveness trials, and in particular, comparative effectiveness trials which compare different intervention dosages or delivery modes, are limited [11]. Furthermore, exercise prescribed in effectiveness trials tends to be of lower intensity than the moderate-to-high intensity that is currently recommended in exercise oncology guidelines [12].

While there is growing momentum behind the campaign to provide exercise to cancer survivors, funding models that support cancer rehabilitation services are highly varied across health systems in developed countries [1]. Even when a government or third-party payer funding model supports exercise services, there are likely to be out-of-pocket expenses for the patient [13]. Few countries have health system funding available to support exercise services, with Australia and Germany being the noted exceptions. The United States and the United Kingdom both have large community-based programs (e.g., Livestrong and MoveMore), however funding for these programs relies on community partnerships, leaving the program vulnerable to loss of support [3]. Australia has arguably one of the most universal funding models, with all cancer patients eligible for reimbursement for up to five sessions with an allied health professional (e.g., accredited exercise physiologist or physiotherapist) per year under a government-funded chronic disease management scheme [14]. However, there is no evidence that this real-world level of contact with an exercise professional is safe or effective [15].

As observed in pharmaceutical and surgical oncology trials [4,5], sample bias and interventions that are not representative of clinical practice limit external validity and present as barriers to research translation. Therefore, the aim of the SAFE trial was to evaluate the safety and feasibility (primary outcomes), and effect (secondary outcomes) of an exercise intervention delivered via low-level versus high-level supervision, in an understudied subgroup of women with breast cancer. Women were randomised to receive the exercise intervention delivered via a supervision-model consistent with either,

real-world: representative of the model currently funded by the Australian healthcare system [14]. This involved five supervised sessions across 12 weeks, with all other exercise sessions unsupervised. This group was the “low-supervision group”.research: consistent with conditions commonly observed in previous exercise oncology trial protocols [10]. This involved 20 supervised sessions across 12 weeks, with all other exercise sessions unsupervised. This group was the “high-supervision group”.

The understudied target population was breast cancer survivors who had a high burden of disease based on disease stage, comorbidities, and persistent treatment-related side-effects, and were insufficiently active. These women are typically excluded from studies or less likely to volunteer [6].

## 2. Methods

### 2.1. Design and Participants

The SAFE trial (ANZ Clinical Trials Registry: ACTRN12616000547448; 2016-18) was a randomized, comparative effectiveness trial. Ethical approval was obtained from Human Research Ethics Committees at the Queensland University of Technology and participating hospitals, and reporting and conduct adhered to the Consolidating Standards of Reporting Clinical Trials (CONSORT) guidelines [16]. Due to recruitment challenges and financial constraints, modifications were made to the protocol after trial registration but prior to enrolling participants. Details of these changes and justifications are summarized in Appendix A.

Study inclusion and exclusion criteria are outlined in Table 1. Potentially eligible women were referred by breast care nurses at one public and two private hospitals, either during routine appointments or via mail-out. Women could also self-refer through study advertisements or the clinical trials registry. All interested women were screened for eligibility. Following written consent (from the participant and treating doctor) and baseline assessment, participants were randomized into either the low-supervision (5 supervised sessions) or high-supervision (20 supervised sessions) intervention groups. Participants were randomized using computer-generated block randomization (blocks of four), stratified by treatment status at baseline (current versus completed treatment, excluding hormone therapy). Allocations were stored in sequentially-numbered envelopes which were given to participants after baseline testing. Staff involved in data collection (but not delivery of the intervention) were blinded to group allocation.

### 2.2. Exercise Intervention

All participants, regardless of group, received an individualised, progressive 12-week exercise prescription, delivered via an ExP during supervised sessions. This exercise intervention was completed during both supervised sessions and independently during unsupervised sessions. The two groups differed only according to the number of sessions supervised by an ExP across the 12-week intervention. Key differences and similarities between the groups are summarized in online Appendix A.

### 2.3. Low-Supervision Group

The low-supervision group were allocated five supervised sessions across the 12-week intervention period. The first session was scheduled during week 1 and scheduling of subsequent sessions was determined jointly by the ExP and the participant. Participants completed exercise during supervised sessions, as well as independently during unsupervised sessions, as prescribed by the ExP.

### 2.4. High-Supervision Group

Participants in the high-supervision group were allocated 20 sessions. In weeks 1–8, participants had two supervised sessions per week and in weeks 9–12 participants had one supervised session per week. As with the low-supervision group, prescribed exercise was to be completed during both supervised and unsupervised sessions.

### 2.5. Supervised Sessions

Supervised sessions involved discussion of weekly exercise prescription, and provision of exercise supervision (ensuring correct technique, monitoring intensity) and exercise counselling (behaviour change and support on overcoming barriers to participation). The ExP used a patient-centered approach during all sessions by following the Chronic Disease Self-Management Intervention Model [17]. This model enables collaborative discussions and consideration of individual circumstances. The ExPs delivering the intervention were all tertiary-trained exercise physiologists with additional study-specific training in exercise oncology. Participants were provided with written exercise prescriptions with details of all exercise prescribed which was to be completed outside of supervised sessions (e.g., type, frequency, intensity, and duration of unsupervised exercise).

### 2.6. Exercise Prescription

The target weekly exercise dosage was based on Australian guidelines at the time of study commencement [18,19] that recommend the equivalent of 150 min of mixed-mode (i.e., aerobic exercise and at least two resistance exercise sessions per week), moderate-intensity exercise (i.e., 600 metabolic-equivalent minutes, MET-mins). In line with the guidelines, there was flexibility in the specific mode of aerobic and resistance exercises included in individual prescriptions. The goal of the exercise program was to support each participant to reach the target exercise dosage each week through exercises that were judged by the ExP as safe and specific to the participant’s goals, while also accommodating the participant’s exercise mode preferences. Participant characteristics and exercise tolerance/capacity, alongside clinical judgement, were used to determine the most appropriate exercise types. The exercise dosage starting point, weekly volume, exercise mode and progression rate for each participant were also considered (for example sessions see Table 2). Exercise intensity was prescribed and monitored using the Rating of Perceived Exertion (RPE, 6–20 scale [20]). Further details on exercise prescription in line with the Consensus on Exercise Reporting Template [21] are in Appendix A.

### 2.7. Data Collection

Personal, diagnostic, and treatment- and health-related characteristics (Table 3) were collected at enrolment via self-report over the telephone.

### 2.8. Primary Outcomes

Safety and feasibility data were collected systematically throughout the intervention. Participants were asked to record the details of all exercise completed (e.g., duration, intensity, mode) and any adverse events (AEs) in a study-specific logbook on a daily basis; these data were then recorded in case management folders by the ExP at each session. Study records of recruitment, retention and delivery were also extracted from the case management folders for feasibility analysis. Objectively-assessed and patient-reported outcomes were assessed at baseline (pre-intervention) and post-intervention (12 weeks post-baseline), with patient-reported outcomes also assessed at 12 weeks post-intervention.

AEs (safety outcome) were defined in accordance with the Good Clinical Practice Guidelines [22] as ‘any unfavorable and unintended sign, symptom or disease that occurs in a participant whether it is considered to be study- or non-study-related’. AEs were deemed to be exercise-related AEs (ExAEs) if they occurred during or within two hours of supervised or unsupervised exercise or had a clear mechanism relating the AE to exercise (as determined by treating medical team or senior ExP). Participants were asked at each supervised session if they had experienced any AEs since the previous session. AE descriptions and severity ratings were based on the Common Terminology Criteria for Adverse Events, Version 4 [23]. Exercise-related AEs could include injuries (including falls), medical events (e.g., unstable angina), and/or exacerbations of treatment-related side effects. Adverse events were classified as grade 3 if they led to hospitalisation and/or led to limitations in self-care activities of daily living [23]. The a priori acceptable threshold for safety was no grade 3 or above ExAEs.

Feasibility was defined as the median volume of weekly exercise completed during the intervention compared with the weekly intervention target (600 MET-mins), similar to the method used by Scott et al. [24]. The intervention was to be deemed feasible if the feasibility rate for the group was ≥75%. MET-mins for each session were calculated as minutes of exercise multiplied by the MET-value equivalent to the reported RPE (conversion of RPE to MET-value was extrapolated from Norton et al. [25]).

### 2.9. Secondary Outcomes

Quality of life, exercise self-efficacy and total weekly PA were assessed via self-report questionnaire using the Patient Reported Outcomes Measurement Information System (PROMIS^®^) Global Health Scale [26], a cancer-specific exercise-barriers self-efficacy scale [27], and Active Australia Survey [28], respectively. Aerobic fitness, upper-body strength and lower-body strength were measured using the 6-min walk test (following the protocol of the American Thoracic Society [29]), the YMCA bench press [30] with a modified weight of 10 kg (to accommodate potential upper-limb post-surgical limitations) and the 30-s sit-to-stand [31], respectively. Clinically-relevant changes were determined a priori based on thresholds identified from previous studies in similar populations or based on distribution of baseline values (½ standard deviation (SD)) when minimally-important differences had not previously been assessed [32,33,34,35].

### 2.10. Statistical Analysis

Safety and feasibility were reported as number (%) and group medians (minimum, maximum) and a Mann-Whitney U-test was used to test significance for differences between groups [24]. Generalised estimating equations were used to determine time, group, (high-supervision versus low-supervision) and time by group effects [36,37,38]. Estimated means, 95% confidence intervals (CI) and *p*-values are reported for each estimate and mean differences. CI and *p*-values from pairwise results are presented in exploration of significant time effects and group by time interactions. The sample size for this study was based on estimating 80% compliance to the exercise target with a 95% confidence interval of ±10%. A group size of 30 participants allowed for 5% withdrawals and 5% loss-to-follow-up across the 24 weeks of the intervention and follow-up period (sample size formula based on Hooper, 2019 [39]). Intention-to-treat principles were applied during data analyses. All analyses were undertaken using IBM SPSS Statistics for Windows, version 25.0, IBM: New York, USA.

## 3. Results

The two exercise groups had similar personal, treatment and behavioural characteristics at baseline (Table 3). Two-thirds of women (*n* = 39) reported at least one comorbidity and 98% (*n* = 59) reported one or more persistent treatment-related side-effects, with an average of 4.3 (SD 2.3) comorbidities and/or side-effects per participant.

### 3.1. Primary Outcome: Safety

There were no grade 3 or above ExAEs (Table 4), and there was no difference in AE rate, type, or severity between groups (Appendix A). The majority of ExAEs were mild (grade 1 *n* = 86/126 ExAEs); 20% required an interruption or modification to the intervention. Two ExAEs occurred during and following baseline testing (*n* = 1, chest pain during the 6-min walk test; *n* = 1, ‘severe’ upper-body delayed-onset muscle soreness in days following testing, likely caused by YMCA bench press test). These were not included in the analysis of intervention safety. The participant with exercise-induced chest pain was referred to her medical team for clearance prior to commencing the intervention.

### 3.2. Primary Outcome: Feasibility

The intervention completion rate was 98% (59/60 women; Figure 1: Flow diagram of recruitment and retention of participants). The median percentage of attended scheduled ExP sessions was 100% (high: median 20 [range 4–20]; low: median 5 [range 4–5]). Median weekly MET-mins for both groups exceeded the feasibility threshold of 75% (i.e., 450 MET-mins) and the weekly goal of 600 MET-mins, although this was higher in the high-supervision group (817 MET-mins [min–max: 446–2103]) compared to the low-supervision group (663 MET-mins [min–max: 30–1924]; *p* = 0.047). There was wide variation within groups in weekly MET-mins observed across the intervention (see Figure 2). Figure 3 provides a graphical representation of the variation in weekly MET-mins between individuals within each group, as well as within each individual. A median of two resistance sessions per week were completed by women in both groups (min–max: High = 0–7; Low = 0–5).

### 3.3. Secondary Outcomes: Effect

Clinically-relevant improvements between baseline and post-intervention for quality of life, PA and fitness were observed for both groups (*p* < 0.05; Table 5). Quality of life improvements were maintained at the 12-week follow-up. Despite a reduction in minutes of PA between post-intervention and follow-up, PA levels at the 12-week follow-up remained higher than baseline (*p* < 0.05). 

Group by time interactions were observed for exercise-barriers self-efficacy and upper-body strength (*p* < 0.05; Table 5), with the high-supervision group showing improvements at post-intervention (gains of 11.3 points and 17.9 repetitions, respectively), which were maintained at follow-up. In contrast, self-efficacy and upper-body strength scores for the low-supervision group remained unchanged over time (1.6 point and 3.6 repetition change between time 1 and 2, respectively, which is below the a priori defined clinically relevant threshold). Sit-to-stand scores for both groups improved between baseline and post-intervention, although only improvement in the High-supervision group met the clinically-relevant threshold.

## 4. Discussion

The SAFE trial was an individualised, 12-week exercise intervention with a weekly target exercise dosage of 600 MET-mins. We hypothesized that despite recruiting a potentially higher-risk cohort, the exercise intervention would be safe and feasible (primary outcomes) and beneficial for both the low-supervision and high-supervision groups. Our findings indicate that this exercise is safe and feasible and improves quality of life, PA, and fitness in breast cancer survivors with additional comorbidities and persistent treatment-related side-effects. These findings were consistent, irrespective of whether the intervention was delivered via 20 or five supervised sessions with an ExP. Nonetheless, clinically-relevant improvements in strength outcomes and self-efficacy were observed between pre- and post-intervention for those in the high-supervision group and not for those in the low-supervision group. Results of this comparative effectiveness trial suggest that the current Australian funding for provision of exercise services for those with cancer provides a valuable foundation for improving the lives of breast cancer survivors. However, additional supervision may contribute to greater and potentially more sustainable benefit.

### 4.1. Safety and Feasibility

No grade 3 or above ExAEs were observed during the intervention period. High average compliance, which exceeded the target exercise dosage, was reported irrespective of group. These findings support individually-prescribed exercise meeting weekly exercise targets recommended to the wider cancer population as safe and feasible for women with more advanced breast cancer, and with additional comorbidities and/or persistent, treatment-related side-effects. Nonetheless, these findings need to be placed in the context from which they are drawn. Specifically, exercise prescriptions were individually tailored by ExPs with tertiary qualifications and oncology experience. All participants (irrespective of group allocation) were routinely educated and advised on exercise safety and on using symptom response to guide subsequent exercise prescription. Although rare, there were reports of breast cancer specific concerns, including lymphoedema, cording and shoulder pain, presenting during the intervention. These are neither unique to SAFE [8] nor do they represent contraindications to exercise participation [2]. However, these issues required sensitivity and consideration in subsequent exercise prescription, highlighting the need for qualified ExPs with oncology-specific training. Half of the AEs reported in SAFE informed a purposeful modification of the subsequent exercise prescription (either mode, frequency, duration, or intensity), and one-quarter of the AEs required referral to other allied health professionals or the treating team. As such, while the SAFE intervention, including when delivered under low-level supervision conditions, was deemed safe, trained ExPs implemented the intervention with input when needed from the wider cancer care team or other allied health professionals.

SAFE AE rates were higher than the AE rates reported following a meta-analysis of studies evaluating exercise involving women with breast cancer [8]. The differences in rates are likely reflective of data collection procedures rather than safety; the majority of trials included in the meta-analysis had poor AE assessment and reporting procedures (34% did not report safety at all). Lack of evidence demonstrating patient safety contributes to clinicians not encouraging exercise participation [40]. While findings from this work can be used to reassure clinicians that exercise is safe, there remains a clear need to improve safety reporting protocols within exercise oncology trials more broadly [41].

Average weekly exercise reported by both SAFE groups exceeded the intervention target. This is consistent with the high feasibility rates reported in exercise studies involving women with stage II or above breast cancer [8] but is notably higher when compared to a study that only included women with stage IV breast cancer undertaking chemotherapy [24]. Specifically, SAFE participants completed an average of 123% of the target exercise dosage, whereas participants in the metastatic breast cancer study only completed 61% of planned MET-hours of exercise. The higher feasibility rates observed in SAFE may be due to the small portion of women with stage IV disease (12%), most (63%) having had completed treatment during study participation, and location of the exercise intervention (SAFE was primarily home-based and not supervised). Nonetheless, the group averages fail to fully reflect individual levels and fluctuations over the intervention duration. For example, most participants in SAFE had at least two weeks in which they completed less than the weekly exercise target (see Figure 3). The impact of non-compliant weeks on patient outcomes and whether a week of low-volume or no exercise can be balanced by a week that exceeds the exercise target without changing outcomes is yet to be determined. Regardless, the individual feasibility results reinforce the need for flexible exercise prescriptions that recognize the likelihood of breast cancer survivors experiencing ‘good’ and ‘bad’ weeks [2]. Conversely, there were a subgroup of women (23% in high, 17% in low) who met the target every week, even in the presence of barriers. It is plausible, but at this point untested, that at least for some of these women, fewer sessions may have reduced participant burden without undue adverse impact on short- or longer-term effect on outcomes, including exercise levels.

### 4.2. Effect Outcomes

The supervision level of the high-supervision group was designed to represent that commonly observed in published breast cancer and exercise trials (i.e., one to two supervised sessions/week) [10,42,43]. In line with previously published findings [10,43], improvements in quality of life and function were observed for those in the high-supervision group immediately at post-intervention and at 12 weeks post-intervention. The similar improvements observed in the low-supervision group for quality of life, PA, and fitness immediately post- and 12-weeks post-intervention indicate that the Australian funding model for allied health services (five sessions) may be an effective platform for exercise therapy delivery following breast cancer, at least when all five sessions are provided over a 12-week period. However, while similar improvements in quality of life and fitness were observed between the high-supervision and low-supervision groups immediately post-intervention, clinically-relevant improvements in strength were observed only in the high-supervision group. Furthermore, the high-supervision group also showed improvements in self-efficacy (the ability to overcome barriers to exercise), whereas those in the low-supervision group did not, and PA levels were higher throughout and beyond the intervention period for those in the high- versus the low-supervision group. Epidemiological evidence which shows that higher levels of lean tissue (which is directly associated with strength), and PA are independently associated with improved survival post-breast cancer [33,44,45] may suggest that the differences between the high- and low-supervision groups are worthy of attention. Furthermore, the potential benefit of exercise is unclear if delivered via a maximum of five sessions over a 12-month period or when the five sessions are shared among other allied health services typically required by breast cancer survivors including physiotherapy, podiatry, and dietetics, which is the intent of the current Australian funding model.

### 4.3. Strengths and Limitations

Strengths of the SAFE trial include the successful recruitment of an understudied sample of breast cancer survivors, comprehensive reporting of safety and feasibility data, and evaluation of a real-world delivery model. Limitations of SAFE include potential reporting bias of AE data (e.g., higher recall bias for those in the low-supervision group than the high-supervision group; although likely small for grade 3 or higher AE over a 12-week period), and a relatively short follow-up (12-weeks post-intervention). Furthermore, most of the sample had a history of at least irregular exercise and, while not sufficiently active on trial commencement, they were also not sedentary. The potential impact of this bias on safety, feasibility and efficacy findings is unclear.

### 4.4. Clinical Implications and Future Research

The results of this trial suggest that an exercise intervention delivered under real-world conditions is appropriate for implementation in a representative cohort of breast cancer survivors. However, future research is needed to better understand whether the greater gains in health outcomes that come with higher levels of supervision translate into longer term quality of life and survival benefits and sustained behaviour change. Future planned research related to this work includes analysis of the cost-effectiveness of the high- versus low-supervision group and exploration of the impact of compliance on patient outcomes, with subsequent findings aiding the translation of these results into clinical practice.

## 5. Conclusions

SAFE provides evidence that exercise prescribed alongside education and support in exercise safety and behaviour change via either five or 20 sessions supervised by an ExP is safe, feasible (primary outcomes) and beneficial (secondary outcomes) for women with stage II+ breast cancer. Individually-targeted exercise is appropriate for previously insufficiently active women with breast cancer, even in the presence of chronic side effects and/or comorbidities. Five sessions with an ExP over a 12-week period led to improvements in quality of life and physical function. However, superior gains were observed for women who had 20 sessions with an ExP over the same duration.

## Figures and Tables

**Figure 1 cancers-14-01528-f001:**
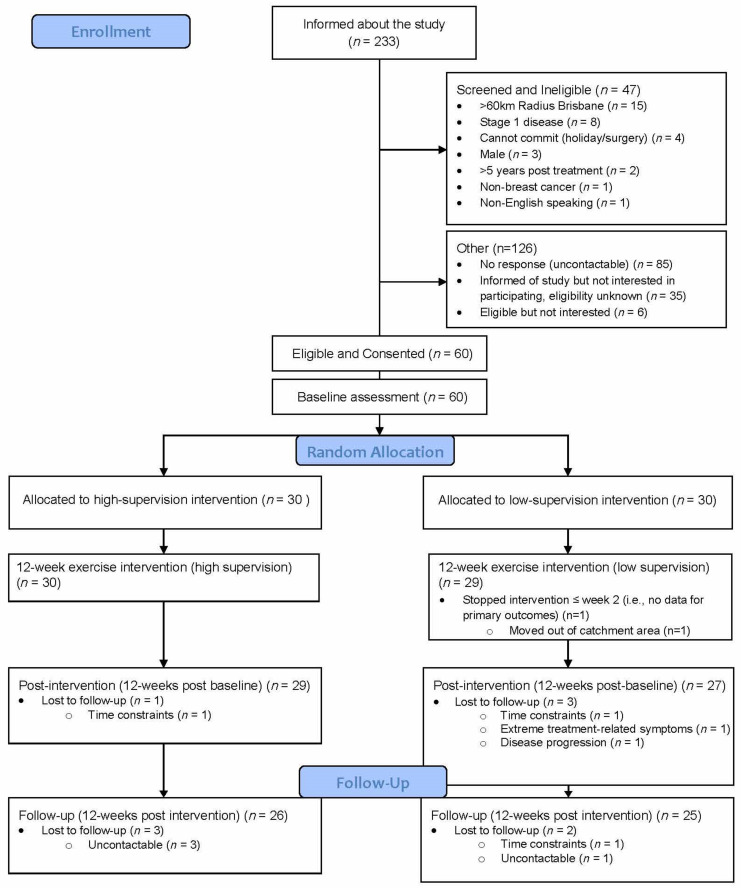
Flow diagram of recruitment and retention of participants.

**Figure 2 cancers-14-01528-f002:**
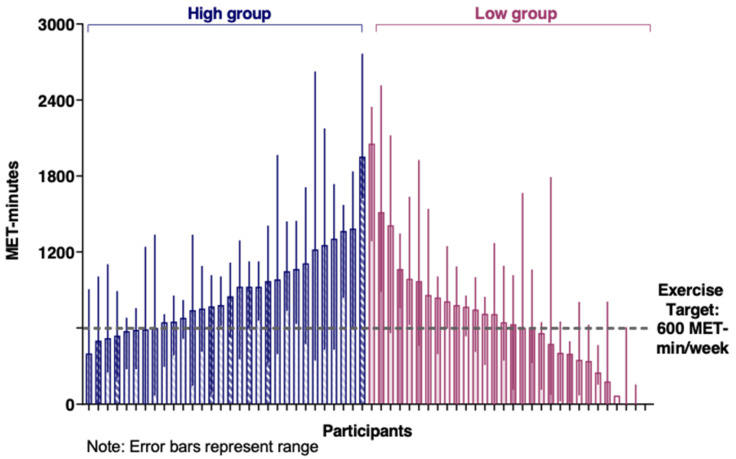
Median weekly exercise volume undertaken per participant in the high-supervision and low-supervision groups.

**Figure 3 cancers-14-01528-f003:**
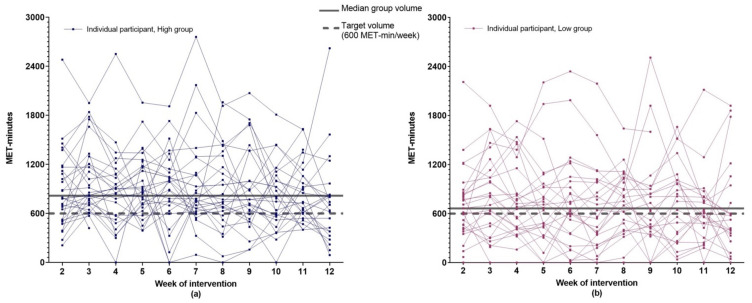
Weekly exercise volume over the 12-week intervention for each participant. (**a**) Data from participants in the high-supervision group; (**b**) Data from participants in the low-supervision group.

**Table 1 cancers-14-01528-t001:** Participant Eligibility Criteria.

Inclusion Criteria		Exclusion Criteria
Must Meet ALL Criteria	Must Meet ≥ 1 Criterion	Meets ANY Criterion
≥18 years of age Resides/works in Greater Brisbane-area Currently undergoing treatment for stage II+ breast cancer, OR completed treatment for stage II+ breast cancer within ≤5 years Currently does not meet national physical activity level recommendations ^a^	Presence of ≥1 comorbidity or chronic disease including hypertension, arthritis, obesity, osteoporosis, type II diabetes Chronic breast cancer treatment sequelae of moderate intensity or higher. Examples include lymphoedema, neuropathy, fatigue, or arthralgia	Planned pregnancy Surgery (e.g., hysterectomy or breast reconstruction) Holidays during intervention period

^a^ Engages in <150 min of self-reported structured exercise per week.

**Table 2 cancers-14-01528-t002:** Example exercise sessions including prescription parameters.

Exercise Prescription Parameter	Aerobic Exercise Prescription	
	Average Participant	Deconditioned Participant; or Participant Experiencing Period of Activity-Limiting Pain or Nausea
Mode of Exercise ^a^	Walk, cycle (stationary or bicycle), swim ^a^	Walking (flat road, treadmill, shopping center) or stationary cycling ^a^
Frequency, sessions ^b^ per week	3–4	6
Intensity	Moderate	Moderate
RPE ^c^, 6–20 Borg Scale	12–14	11–13
Duration, minutes		
Individual session	20–40	20 (broken into shorter bouts, as needed, throughout the day, e.g., 4 × 5 min)
Total weekly	110	120
Eliciting progressing overload		
Recommendations	Increase speed, load, or incline to maintain RPE	Increase duration of bouts until able to complete 20 min continuously
Example	Increase pace, include hills or inclines, intervals of higher speed to maintain overall intensity target across session	Use “talk-test” to identify threshold for moderate-intensity. Symptoms (e.g., fatigue, pain) may influence RPE more than cardiovascular response
**Exercise Prescription Parameter**	**Resistance Exercise Prescription**	
	**Average Participant**	**Deconditioned Participant; or Participant Experiencing Period of Activity-Limiting Pain or Nausea**
Frequency, sessions ^b^ per week	2	2
Intensity	Moderate	Moderate
RPE ^c^, 6–20 Borg Scale	12–14	11–13
Repetitions in reserve	Aim for 2–3 repetitions in reserve at the end of each set	Aim for 3–4 repetitions in reserve at the end of each set
Duration, minutes		
Individual Session	20	15
Total Weekly	40	30
Session components		
Focus	Muscular strength	Muscular endurance
Repetition range	8–12	15–20+
Set range	2–3	2
Example home-based resistance exercises ^a^		
Lower body	Squat Calf-raise on step including dorsiflexion Lateral banded walk Resistance band deadlift	Sit-to-stand Supine bridge Side-lying hip abduction
Upper body	Bent-over row (single-arm, dumbbell)	Resistance band row Resistance band chest press
Exercise Recommendations	4–5 major muscle group exercises, 1 targeted exercise (functional or injury-specific where required)	
Eliciting progressive overload	Increase reps or sets, increase resistance or weight, alter exercise tempo	

^a^ This is not an exhaustive list; if a participant wanted to engage in other aerobic or resistance-based exercises the exercise professional would include the activity in the prescription if it was deemed safe and appropriate to the participant’s goals (e.g., dragon boat training, gym classes, boxing, machine-based resistance exercises). ^b^ These exercise sessions may have been completed unsupervised or incorporated as part of one of the supervised sessions. ^c^ RPE: Rating of Perceived Exertion.

**Table 3 cancers-14-01528-t003:** Baseline Characteristics.

	Total *n* = 60	High Group *n* = 30	Low Group *n* = 30
Personal Characteristics	Mean (SD) or *n* (%)		
Age (years), mean (SD)	50.1 (9.0)	51.0 (9.5)	49.2 (8.5)
<50 years	29 (48%)	16 (53%)	13 (43%)
≥50 years	31 (52%)	14 (47%)	17 (57%)
Household income			
Lower income (bottom 40th percentiles) ^a^	24 (40%)	12 (40%)	12 (40%)
Marital status			
Married/de-factor	42 (70%)	22 (73%)	20 (67%)
Private Health Insurance (Yes)	48 (80%)	25 (83%)	23 (77%)
Body-mass index (kg/m^2^)	28.9 (6.2)	29.2 (6.7)	28.6 (5.7)
Body-mass index (*n*, %)			
Healthy or underweight	19 (32%)	9 (30%)	10 (33%)
Overweight	18 (30%)	9 (30%)	9 (30%)
Obese	23 (38%)	12 (40%)	11 (37%)
Minutes of structured exercise/week, mean (SD)	42.0 (57.6)	41.0 (55.8)	41.0 (60.1)
**Diagnostic Characteristics**	**Median (range) or *n* (%)**		
Breast Cancer stage			
Stage II	28 (47%)	12 (40%)	16 (54%)
Stage III	20 (33%)	14 (47%)	6 (20%)
Stage IV	7 (12%)	3 (10%)	4 (13%)
Unsure or unknown ^b^	5 (8%)	1 (3%)	4 (13%)
Months since diagnosis ^c^	18 (2–243)	16 (2–215)	24 (2–243)
Side of breast cancer			
Dominant side	27 (45%)	12 (40%)	15 (50%)
Non-dominant side	30 (50%)	17 (57%)	13 (43%)
Bilateral	3 (5%)	1 (3%)	2 (7%)
**Treatment Characteristics**	***n* (%)**		
Most extensive surgery			
Mastectomy	43 (72%)	19 (64%)	24 (80%)
Lumpectomy	16 (27%)	10 (33%)	6 (20%)
No surgery	1 (2%)	1 (3%)	0 (0%)
No. of nodes removed			
0	0 (0%)	0 (0%)	0 (0%)
1–4	17 (28%)	7 (23%)	10 (33%)
5–9	6 (10%)	2 (7%)	4 (13%)
10+	25 (42%)	13 (43%)	12 (40%)
Unsure or unknown	12 (20%)	8 (27%)	4 (13%)
Treatment status			
Currently receiving treatment ^c, d^	22 (37%)	10 (33%)	12 (40%)
Treatments received (current or past)			
Chemotherapy (yes)	55 (92%)	25 (83%)	30 (100%)
Radiation therapy (yes)	45 (75%)	25 (83%)	20 (67%)
Hormone therapy (yes)	32 (53%)	18 (60%)	14 (47%)
**Health Characteristics**	***n* (%)**		
Number of comorbidities	1 (0–6)	1 (0–6)	1 (0–5)
0 comorbidities	21 (35%)	9 (30%)	12 (40%)
1–2 comorbidities	29 (48%)	15 (50%)	14 (47%)
3–4 comorbidities	7 (12%)	4 (13%)	3 (10%)
5–6 comorbidities	3 (5%)	2 (7%)	1 (3%)
Number of side effects	4.4 (2.1)	4.7 (2.1)	4.1 (2.1)
0–2 side effects	11 (19%)	5 (17%)	6 (20%)
3–4 side effects	24 (40%)	10 (33%)	14 (46%)
5–6 side effects	15 (25%)	10 (33%)	5 (16%)
7+ side effects	10 (17%)	5 (17%)	5 (16%)
Number of side effects (≥moderate severity)	2.8 (1.9)	3.1 (1.9)	2.5 (1.8)
0–2 moderate+ side effects	28 (47%)	12 (40%)	16 (54%)
3–4 moderate+ side effects	21 (35%)	11 (37%)	10 (33%)
5–6 moderate+ side effects	9 (15%)	6 (20%)	3 (10%)
7+ moderate+ side effects	2 (3%)	1 (3%)	1 (3%)

^a^ Based on cut-off for lower-income households, i.e., lowest 40th percentiles of gross household income (ABS Survey of Income and Housing, 2015–2016 and 2017–2018). Data missing for *n* = 10; ^b^ Confirmed stage II or above based on referral by medical team; ^c^ Not including hormone therapy (i.e., women undergoing just hormone therapy were classified as having completed treatment); ^d^ Stratification factor (currently undergoing treatment: yes, no).

**Table 4 cancers-14-01528-t004:** Safety of exercise: Adverse events during the SAFE intervention.

	All Women	High Group	Low Group
	*n* = 59	*n* = 30	*n* = 29
**Primary Outcome:**			
Number of ≥ grade 3 exercise-related AE	0	0	0
**Adverse Events**			
Number of AEs (total)	177	136	41
Number of women reporting AEs	41 (69%)	23 (77%)	18 (62%)
Median (range) number of AEs per participant	2 (0–19)	4 (0–19)	1 (0–7)
**Exercise-related Adverse Events**			
Number of exercise-related AEs (total)	126	103	23
Number of women reporting exercise-related AEs	34 (58%)	23 (77%)	11 (38%)
Median (range) number of exercise-related AEs per participants	1 (0–14)	3 (0–14)	0 (0–6)
Number of women reporting exercise-related AEs			
0 AEs	26 (44%)	7 (23%)	19 (66%)
1–2 AEs	16 (27%)	8 (27%)	8 (28%)
3–4 AEs	8 (14%)	6 (20%)	2 (7%)
5–10 AEs	8 (14%)	7 (23%)	1 (3%)
>10 AEs	2 (3%)	2 (7%)	0 (0%)

AE: Adverse Event.

**Table 5 cancers-14-01528-t005:** Efficacy of exercise: Mean and 95% confidence intervals of health outcomes at baseline, post-intervention and follow-up assessments.

	**Baseline (T1)**		**Post-Intervention (T2)**		**Follow-Up (T3)**		**Δ T1 to T2 ^a^**	**Δ T2 to T3 ^a^**	**Δ T1 to T3 ^a^**
	*Mean*	*95% CI*	*Mean*	*95% CI*	*Mean*	*95% CI*	M_diff_ (95% CI)		
PROMIS Global Physical Health							5.91 (4.2–7.6) ^c,d^	0.10 (−1.6–1.7)	6.01 (3.9–8.0) ^c,d^
High	40.4	(38.4–42.5)	46.7	(44.2–49.3)	47.1	(44.0–50.2)			
Low	40.6	(37.6–43.5)	46.1	(43.6–48.5)	45.8	(42.9–48.8)			
			***GEE ^b^: Group p =* 0.71*; Time p < *0.01*; Group × Time*** ***p* = 0.79**						
PROMIS Global Mental Health							5.4 (3.6–7.0) ^c,d^	−0.8 (−3.0–1.4)	4.5 (2.0–7.0) ^c^
High	41.6	(39.0–44.3)	48.4	(45.8–51.1)	47.0	(44.0–50.1)			
Low	41.9	(39.3–44.5)	45.8	(43.1–48.5)	45.5	(43.0–48.0)			
			***GEE ^b^: Group p = *0.37*; Time p < *0.01*; Group × Time*** ***p* = 0.25**						
Exercise-barrier self-efficacy									
High	35.7	(29.1–42.2)	47.0	(41.0–53.0)	49.2	(40.9–57.5)	11.3 (4.5–18.1) ^c,d^	2.2 (−5.1–9.6)	13.5 (3.9–23.3) ^c,d^
Low	32.1	(25.7–38.5)	33.7	(27.5–40.0)	29.8	(22.5–37.2)	1.6 (−4.4–7.7)	−3.9 (−9.3–1.5)	−2.3 (−8.4–3.9)
			***GEE ^b^: Group p = *0.002*; Time p =* 0.02*; Group × Time p =* 0.02 **						
Physical Activity ^e^							244.1 (172.7–315.5) ^c,d^	−31.4 (−124.0–61.3) ^d^	212.7 (120.1–305.4) ^c,d^
High	93.1	(70.0–116.2)	381.8	(289.5–474.1)	349.2	(224.2–473.5)			
Low	140.7	(86.1–195.3)	340.2	(249.2–431.1)	310.0	(190.6–429.4)			
			***GEE ^b^: Group p =* 0.79*; Time p <* 0.01*; Group × Time*** ***p =* 0.42**						
6-min walk test							53.6 (35.7–71.4) ^c,d^	- ^f^	- ^f^
High	494.0	(454.0–530.0)	547.0	(518.0–576.0)	- ^f^				
Low	510.0	(479.0–541.0)	563.0	(541.0–585.0)	- ^f^				
			***GEE ^b^: Group p =* 0.39*; Time p <* 0.01*; Group × Time*** ***p =* 0.90**						
Modified-YMCA bench press								- ^f^	- ^f^
High	27.3	(18.0–36.7)	43.5	(32.6–54.4)	- ^f^		17.9 (5.5–30.3) ^c,d^		
Low	25.6	(19.6–31.6)	29.1	(21.6–36.7)	- ^f^		3.6 (−0.3–7.4)		
			***GEE ^b^: Group p =* 0.18*; Time p < *0.01*; Group × Time*** ***p <* 0.01**						
30-s sit to stand								- ^f^	- ^f^
High	11.6	(10.0–13.1)	15.0	(13.2–16.8)	- ^f^		3.4 (1.7–5.1) ^c,d^		
Low	11.3	(10.1–12.5)	12.6	(11.4–13.8)	- ^f^		1.3 (0.2–2.5) ^c^		
			***GEE ^b^: Group p = *0.14*; Time p <* 0.01*; Group × Time*** ***p =* 0.05**						

T1: Baseline (pre-intervention); T2: Post-intervention (12 weeks post-baseline); T3: 12-week follow-up (12 weeks post-intervention). ^a^ Change scores are reported for whole cohort if no significant group by time interaction (Generalized Estimating Equations (GEE) group ***×*** time *p* > 0.05); reported by group if significant group by time interaction (GEE group ***×*** time *p* ≤ 0.05); ^b^
*p*-value derived from Generalised-estimating equation model; ^c^ Statistically significant change between time points *p* < 0.05; ^d^ Clinically relevant/minimally important difference. Defined as a change of: ≥50 m walked in the 6-min walk test [32], ≥20 min of total physical activity per week [33], ≥five units in the PROMIS global physical and mental health scales [34], ≥two repetitions of the sit-to-stand [35], and a change of nine units and 11 repetitions for exercise-barriers self-efficacy and bench press, respectively.; ^e^ Physical activity: Total physical activity as measured by Active Australia Survey (self-report minutes walking + moderate physical activity+ [2 ***×*** vigorous physical activity]); ^f^ 6-min walk test, Bench press and Sit-to-stand were measured at T1 and T2 only.

## Data Availability

De-identified data, dependent variables and participant characteristics may be available for reasonable research purposes via individual request to the authors.

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
