# Peer review of "A Randomised, Comparative, Effectiveness Trial Evaluating Low- versus High-Level Supervision of an Exercise Intervention for Women with Breast Cancer: The SAFE Trial"

_cancers, 2022, doi:10.3390/cancers14061528_

Round 1

Reviewer 1 Report

The topic and findings of the present paper are relevant.

The simple summary and abstract need to be optimized. In my opinion, methods must be more clearly and exhaustively presented/described. You must explain all concepts when they are first presented in the manuscript. Methods must be reproductible. More references should be cited in introduction and Discussion.

In my opinion, this paper present lack of clarity. It is difficult to understand all concepts and methods.

Simple Summary

- Please clear define de the concepts of “high-  versus low-level supervision” in abstract.

- What is a highly supervised condition?

- How are differentiated the high and low levels? Please give more details.

Abstract

  • Please clear define the “low-level supervision” and “high-level supervision”. The characteristics of the low-level supervision group are (…). The characteristics of the high-level supervision group are (…).
  • Please clear present the number of participants in the “low-level supervision” and in the “high-level supervision” groups.
  • Why “all participated in a 12-week, individually-tailored exercise intervention”? This participation was before or after the inclusion in the low- or high-level groups?
  • “primary outcomes were intervention safety (number, type, and severity of adverse events”; adverse events of medicines? Please give more information about the primary outcome.
  • In my opinion, the abstract is not clear. Please improve the compressibility of the abstract.

Keywords: please cite some MeSH terms. https://www.ncbi.nlm.nih.gov/mesh/

- Please define all concepts when they are first presented in the text. In general, the text has too many explanations between (…).

Introduction

  • Usually, an introduction comprises at least 15 to 20 references.
  • Please give more details about international guidelines on the present topic.
  • Please cite more references in introduction. For instance, some systematic reviews and/or meta-analysis is similar or related topics.
  • Please see Please see https://clinicaltrials.gov/ct2/results?cond=Breast+Cancer&term=exercise%2C+physical+activity&cntry=&state=&city=&dist=
  • Clear explain the concepts “High-supervision group” and “Low-supervision group”
  • “Australian healthcare system (i.e., maximum of five supervised sessions per year).” References are missing. How Australian healthcare system regarding supervised sessions? Please give more details. Please note that the present paper is for an international audience.
  • Please clear present the primary and secondary outcomes of trial at the end of the introduction.

  1. Methods

- Who developed the design of the present trial? How was it validated? Where consensus techniques applied? Please give more details.

- Why this design is appropriate? Please give more details.

- Why this trial is relevant? Have you consulted trials on similar or related topics? Please see https://clinicaltrials.gov/ct2/results?cond=Breast+Cancer&term=exercise%2C+physical+activity&cntry=&state=&city=&dist=

- Exercise Intervention and supervision models: Please create two subheadings: one subheading about the “low-level supervision” group and other subheading about the “high-level supervision” group. The characteristics of both groups must be clear and independently presented. These differences may be presented through a tabular format.

- You may create a subheading about the common features between both groups.

- Please create a section about primary and secondary outcomes, please clear present the methods for the primary and secondary outcomes. Definitions in the section of “Data collection” may be here integrated. Please, clear present methods for Safety, Feasibility, and Effect. Please note that the presentation of results should follow the structure of methods.

- “Number of Exercise-related AEs”; What is an “Exercise-related AEs”? How were “Exercise-related AEs” evaluated?

Statistical Analysis

  • Please cite references of similar or related studies that apply similar statistical analysis
  • Please correctly cite SPSS: https://www.ibm.com/support/pages/how-cite-ibm-spss-statistics-or-earlier-versions-spss

Results

  • Table 3: please also present %s; Please do not use symbols, such as “/” or # in Tables.

Discussion

  • Please cite more updated references and discuss more studies.
  • Please discuss more findings from similar or related studies.
  • Please discuss the findings of concluded trials on similar and/or related topics. Please see https://clinicaltrials.gov/ct2/results?cond=Breast+Cancer&term=exercise%2C+physical+activity&cntry=&state=&city=&dist=
  • Please create a subsection in practical implications and future research.
  • Please include in study strengths: What this study adds to the state-of the art?

  1. Conclusion

Conclusion should reply to study objectives. Please place practical implications and future research in discussion.

Limitations

  • Please discuss potential study biases.

  • Please check the format of Tables, Figures and References in the instruction for authors. Please see published papers of this journal to check formats.

Reviewer 2 Report

The manuscript by Spence et al. evaluated the safety, feasibility, and effectiveness of high- versus low-level supervision of exercise intervention focusing on breast cancer patients.

The following comments need to be addressed satisfactorily to enhance the quality and visibility of the manuscript.

  1. A separate table, particularly in the introduction, is necessary, which indicates inclusion and exclusion criteria of the patients, current available information is vague and unclear.
  2. What exactly is High- versus Low-supervision group is unclear; specific information, including an illustration or table, is needed.
  3. Any information on similar studies or clinical trials on complex breast cancer patients should round up the discussion.
  4. The flow chart in Figure 1 is unclear and not appropriately illustrated.
  5. Several long and lengthy sentences need to break and made simple so that it would be easy for breast cancer non-expert readers of the journal.

Round 2

Reviewer 1 Report

- The quality of the paper has been improved, congratulations. Many thanks for all improvements. The topic is relevant for publication.

Some minor comments:

  • Please give some real examples of “exercise-related adverse events” (serious and non-serious) at least in abstract, and methods.
  • Keywords: Exercise; Neoplasms; survivorship; safety; patient compliance; Why some keywords are capitalized (and others not)? Please check in instructions for authors.
  • Introduction: please place study aim at the end of introduction, i.e., “to evaluate the safety, feasibility and effect of an exercise intervention delivered via low-level versus high-level supervision.”
  • Methods: “Data collection Baseline characteristics were collected via self-report at enrolment.” Please give more details. A self-administered questionnaire? Please describe the baseline characteristics (or cite a Table).
  • Statistical Analysis: please cite one or two studies applying similar (or equal) statistical methodologies.
  • Please place the subheading “Primary outcome: Safety” at the beginning of page 10.
  • Lines 290-291: “The intervention completion rate was 98% (59/60 women; Figure 1: CONSORT flow chart)”: the references of CONSORT flow chart are missing.
  • The graphics of Figure 3 are confused; please consider the creation of other more readable graphics (or Tables).
  • Page 15 is blank.
  • Discussion: please create an explicit sentence about the study hypothesis, i.e. “We hypothesised that despite recruiting a potentially higher-risk cohort, the exercise intervention would be safe, feasible (primary outcomes) and beneficial (secondary outcomes) for both the low-supervision and high-supervision groups.”
  • Conclusion: please include the words primary and secondary outcomes in conclusion.

Thanks, from Portugal, Lisbon!
